# Mani-GS: Gaussian Splatting Manipulation with Triangular Mesh

## Abstract

Neural 3D representations such as Neural Radiance Fields (NeRF), excel at producing photo-realistic rendering results but lack the flexibility for manipulation and editing which is crucial for content creation. Previous works have attempted to address this issue by deforming a NeRF in canonical space or manipulating the radiance field based on an explicit mesh. However, manipulating NeRF is not highly controllable and requires a long training and inference time. With the emergence of 3D Gaussian Splatting (3DGS), extremely high-fidelity novel view synthesis can be achieved using an explicit point-based 3D representation with much faster training and rendering speed. However, there is still a lack of effective means to manipulate 3DGS freely while maintaining rendering quality. In this work, we aim to tackle the challenge of achieving manipulable photo-realistic rendering. We propose to utilize a triangular mesh to manipulate 3DGS directly with self-adaptation. This approach reduces the need to design various algorithms for different types of Gaussian manipulation. By utilizing a triangle shape-aware Gaussian binding and adapting method, we can achieve 3DGS manipulation and preserve high-fidelity rendering. Our approach is capable of handling large deformations, local manipulations, and soft body simulations while keeping high-quality rendering. Furthermore, we demonstrate that our method is also effective with inaccurate meshes extracted from 3DGS. Experiments conducted demonstrate the effectiveness of our method and its superiority over baseline approaches. Project page here: https://mani3dgs.github.io/

## 1 Introduction

Manipulating and editing 3D content is essential for content creation and has various applications in movies, gaming, and virtual/augmented reality. 3D model editing enables users to create and modify models flexibly, thereby enhancing production efficiency. The traditional pipeline for modeling and editing a 3D asset with photo-realistic rendering involves a process with geometry modeling, texturing, UV mapping, lighting, and rendering, which is a tedious and time-consuming flow requiring lots of manual work.

Over the past few years, the neural radiance field (NeRF) (Mildenhall et al., 2021) has been widely studied due to its high capability and simple reconstruction process in 3D representation. However, the implicit representation poses challenges for editing.

To address this, some methods are proposed to edit this implicit neural radiance field (Mildenhall et al., 2021). NeRF-Editing (Yuan et al., 2022) is the first to utilize the triangular mesh to help edit the implicit radiance field. They train a canonical NeuS (Wang et al., 2021) and extract the triangular mesh from NeuS (Wang et al., 2021). A tetrahedra grid is then constructed to contain the object mesh. To render the deformed object by editing the triangular mesh, a volume rendering is conducted in the deformed space. The sampling points in deformed space are mapped to canonical space based on the constructed tetrahedra grid, where the points in deformed space have the same tetrahedron barycentric coordinate in the same tetrahedron with their corresponding points in canonical space. Moreover, (Jambon et al., 2023; Liu et al., 2023) demonstrate similar ideas by employing tetrahedra to deform sampling points and achieve editable nerf-based scenes. Instead, Neu-Mesh (Yang et al., 2022) and SERF (Zhou et al., 2023a) define the neural radiance field by associating each mesh vertex with radiance and geometry features. Then they can conduct volume rendering like Point-NeRF (Xu

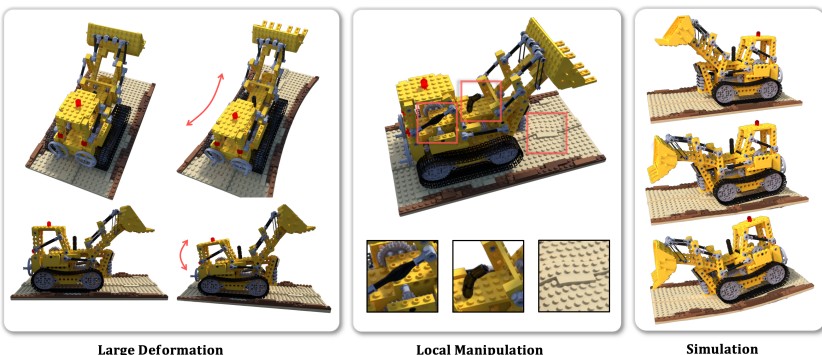

**Large Deformation**    **Local Manipulation**    **Simulation**

Figure 1: Our proposed approach allows for 3DGS manipulation, including *large deformation*, *local manipulation*, and even *physical simulation* (such as soft body simulation), while maintaining high-quality rendering. Please zoom in for more details.

et al., 2022) in deformed space without backward mapping. However, these editing methods based on implicit neural radiance fields still suffer from inconvenient manipulation, suboptimal rendering results, and long training and rendering times.

Meanwhile, 3DGS (Kerbl et al., 2023) has gained significant attention in differential rendering due to its high-fidelity and fast rendering proficiency. However, despite being an explicit 3D representation, it still lacks an effective method for manipulating 3DGS while maintaining high-quality rendering. SuGaR (Guédon & Lepetit, 2023), the work most closely related to our objective, develops a novel algorithm to extract a triangular mesh from 3DGS. Although their main goal is not to facilitate editable photorealistic rendering, they bind the 3DGS to the extracted mesh, enabling model animation as demonstrated in their demo.

This work proposes a method that enables 3DGS manipulation, achieving high-quality and photorealistic rendering. Our key insight is to manipulate 3DGS using a triangular mesh as the proxy, which allows for direct transfer of mesh manipulation to 3DGS with 3DGS self-adaptation. With our methods, we can achieve *large deformation*, *local manipulation*, and *soft body simulaton* with high-quality results as shown in Figure 1, which also avoid the need to design various algorithms for different types of manipulation.

To achieve controllable 3DGS manipulation through the mesh, an intuitive approach is to bind the GS to lay perfectly on the triangle and enforce the GS to be thin enough. After mesh manipulation, the GS will automatically adapt its rotation and position with the attached triangle, as employed in SuGaR (Guédon & Lepetit, 2023). However, SuGaR heavily relies on the accuracy of mesh geometry, inheriting the defects of mesh rendering. Specifically, for inaccurate parts, SuGaR cannot inpaint the missing parts or remove the redundant parts in the final rendering.

Adding an offset to the position of attached Gaussians during the reconstruction may seem like a reasonable solution to compensate for mesh inaccuracy. However, this fixed offset cannot be generalized well to the deformed space after novel manipulation. Our proposed solution is to define a local coordinate system for each triangle, which we refer to as *local triangle space*. We then bind Gaussians to each triangle and optimize the Gaussian attributes, including rotation, position, and scaling, in the attached *local triangle space*.

During mesh manipulation, the attributes in the local triangle space remain unchanged, while the global Gaussian position, scaling, and rotation will be self-adaptively adjusted according to our proposed formula. As a result, our proposed approach enables us to manipulate 3DGS using a triangular mesh while maintaining rendering quality. Since our Gaussians are set to be free outside the triangle, we can also support high-fidelity manipulation even when the Gaussians are bound to an inaccurate mesh, exhibiting a high tolerance for mesh accuracy.

GaMeS (Gao et al., 2024) and Mesh-GS (Waczyńska et al., 2024) are two concurrent works that also employ triangular meshes for Gaussian Splatting manipulation. In particular, GaMeS (Gao et al., 2024) constrains the Gaussians on the surface exactly; Mesh-GS (Waczyńska et al., 2024) permits an offset along the normal direction without adapting the Gaussian scale when the triangle shape

changes. In contrast, our model allows for Gaussian move in the triangle local space which means we can achieve high-quality rendering without the need for accurate mesh. Gaussian scaling also adapts in response to changes in triangle shape when large deformations are applied to the mesh.

In summary, the contributions of our paper are listed as follows:

- We propose a 3DGS manipulation method that can effectively transfer the triangular mesh manipulation to 3DGS and maintain high-quality rendering.
- We introduce a triangle shape aware Gaussian binding strategy with self-adaption, which has a high tolerance for mesh accuracy and supports various 3DGS manipulations.
- We evaluate our method and achieve state-of-the-art results, demonstrating various 3DGS manipulations, including *large deformation*, *local manipulation*, and *soft body simulation*.

## 2 RELATED WORK

### 2.1 NeRF EDITING

Recently, neural radiance field (Mildenhall et al., 2021) (NeRF) has garnered significant attention due to its high-quality and photo-realistic rendering results for novel view synthesis. NeRF represents the scene as a continuous function that maps a spatial location and viewing direction to a volume density and color, which is parameterized by a multilayer perceptron (MLP). Owing to the implicit representation that encodes the scene within the network parameters, editing and deforming the geometry of the NeRF scene explicitly like mesh can be challenging. To enable user editing of NeRF, (Liu et al., 2021) introduce editing conditional radiance fields trained on a shape category. However, it only supports basic editing operations, such as removing/adding object parts or shape transfer. CLIP-NeRF (Wang et al., 2022) achieves NeRF editing with text or images by leveraging CLIP model (Radford et al., 2021) but still can not edit the geometry locally. Some other work (Xiang et al., 2021; Wang et al., 2023b; Bao et al., 2023; Zhan et al., 2023) edit the NeRF in texture level which is not the focus of this paper.

To edit and deform NeRF locally, (Jambon et al., 2023; Yuan et al., 2022; Liu et al., 2023) construct a tetrahedra grid based on the underlying 3D shape. After explicitly deforming the tetrahedra into the posed space for editing, the sampled 3D points are mapped from the posed space to the canonical space through the unaltered tetrahedron, which means the canonical position can be calculated from the shared barycentric coordinate for both deformed and canonical tetrahedron. The density and radiance in the posed space can be calculated for the mapped sampling points in canonical space. On the other hand, (Wang et al., 2023a; Zhou et al., 2023a; Yang et al., 2022) employ mesh as the guidance for deformation. NeuMesh (Yang et al., 2022) presents a novel representation to encode neural implicit field on a mesh-based scaffold for geometry and texture editing. (Wang et al., 2023a) achieves the manipulation of both the geometry and color of neural implicit fields through differentiable colored meshes. Furthermore, there are several methods (Sun et al., 2022; Ma et al., 2022; Zhang et al., 2022; Chen et al., 2023a; Zheng et al., 2022) that particularly focus on editing NeRF for avatar. In this work, we introduce an editing method based on 3DGS for general objects.

### 2.2 MESH-BASED NeRF RENDERING

NeRFs have shown impressive rendering results, however, rendering one pixel using NeRF representation necessitates a volumetric rendering algorithm that involves inferring MLP hundreds of times to estimate their radiance and density. This process is significantly slower than traditional mesh rendering. To accelerate NeRF rendering, several methods (Chen et al., 2023b; Rakotosaona et al., 2023; Yariv et al., 2023; Chen et al., 2023b; Yao et al., 2022; Tang et al., 2023b) have been proposed to combine NeRF representation with mesh reconstruction. By converting this implicit representation to an explicit mesh, these methods may also facilitate applications like editing. In particular, MobileNeRF (Chen et al., 2023b) represents NeRF as a collection of polygons with deep feature textures, which can be rendered using the classic polygon rasterization pipeline, generating a feature vector for each pixel and passing it to an MLP to decode the color. MobileNeRF is capable of achieving rendering even on standard mobile devices. Additionally, NeRFMeshing (Rakotosaona et al., 2023) distills the reconstructed NeRF into a signed surface approximation network to extract 3D mesh and shows the simulation results for editing. BakedSDF (Yariv et al., 2023) proposes a

neural surface-volume representation to extract the mesh and supports editing like material decomposition, appearance editing, and physics simulation. However, the editing results of these methods heavily rely on the accuracy of the extracted mesh. Artifacts present in the mesh will directly influence the editing results. In contrast, our method demonstrates robustness to the reconstructed mesh and can still yield promising results even with the inaccurate mesh.

### 2.3 Gaussian Splatting Editing, Simulation and Animation

3D Gaussian Splatting (Kerbl et al., 2023) presents an innovative 3D Gaussian scene representation, accompanied by a differentiable renderer that attains real-time rendering of radiance fields while maintaining high quality. Initially, 3D Gaussian Splatting focuses solely on static scenes, which has been extended to model dynamic scenes (Wu et al., 2023; Huang et al., 2023; Yang et al., 2023; Lin et al., 2023; Zhou et al., 2023b), human avatars (Zielonka et al., 2023; Qian et al., 2023; Yuan et al., 2023; Hu et al., 2023; Xu et al., 2023; Kirschstein et al., 2023), Gaussian Splatting simulation and animation (Guédon & Lepetit, 2023; Xie et al., 2023; Jiang et al., 2024; Feng et al., 2024). Specifically, SuGaR (Guédon & Lepetit, 2023) proposes a method to extract meshes from 3D Gaussian Splatting with additional regularization, in which they bind 3DGS on extracted mesh and animated with Gaussian Splatting rendering. GSP (Feng et al., 2024) incorporates physically-based fluid dynamics in 3DGS and PhysGaussian (Xie et al., 2023) introduces a unified simulation-rendering pipeline that generates physics-based dynamics with photorealistic renderings. VR-GS (Jiang et al., 2024) achieves interactive physics-based editing in Virtual Reality. In this work, we also introduce a Gaussian Splatting manipulation method that binds Gaussian Splatting to the mesh, achieving state-of-the-art results.

## 3 Method

Recently, due to its exceptional high-fidelity rendering capabilities and fast rendering speed, 3DGS (Kerbl et al., 2023) has emerged as a popular 3D representation in the differential rendering field. However, despite being an explicit 3D representation, it still lacks a way for manipulating this 3D representation for editing while maintaining high-quality rendering after the manipulation. In this work, giving multi-view RGB images of an object as input, we introduce a method for object manipulation that can achieve photorealistic editable rendering by employing Gaussian Splatting.

The pipeline of our method is illustrated in Figure 2 and consists of three main stages. First, we extract a mesh from 3D Gaussian Splatting (3DGS) or a neural surface field for subsequent 3D Gaussian binding (Sec. 3.2). Next, we devise a novel Mesh-Gaussian binding method dedicated to manipulating Gaussian Splatting while maintaining photo-realistic rendering quality (Sec. 3.3). Finally, we describe the types of Gaussian manipulation we support(Sec. 3.4).

### 3.1 Preliminary

Thanks to its superior capability for high-fidelity and rapid rendering, 3D Gaussian Splatting (3DGS) (Kerbl et al., 2023) has recently emerged as a popular 3D representation in differential rendering. 3DGS utilizes explicit 3D Gaussians as its primary rendering primitives. A 3D Gaussian point is mathematically defined as:

$$G(\boldsymbol{x}) = exp(-\frac{1}{2}(\boldsymbol{x} - \boldsymbol{\mu})^\top \Sigma^{-1}(\boldsymbol{x} - \boldsymbol{\mu})). \tag{1}$$

Each 3D Gaussian point is characterized by a 3D mean position coordinate $\boldsymbol{\mu}$ and a covariance matrix $\Sigma$. Additionally, each Gaussian has an opacity $\boldsymbol{o}$ and a view-dependent color $\boldsymbol{c}$ represented by a set of spherical harmonics (SH). To ensure that the covariance matrix $\Sigma$ retains its meaningful interpretation, it is parameterized as a unit quaternion $\boldsymbol{q}$ and a 3D scaling vector $\boldsymbol{s}$, defined as $\Sigma = \boldsymbol{R}\boldsymbol{S}\boldsymbol{S}^\top\boldsymbol{R}^\top$.

To render an image from a specific viewpoint, 3D Gaussians are projected onto the image plane, resulting in 2D Gaussians. The 2D covariance matrix is approximated as:

$$\Sigma' = \boldsymbol{J}\boldsymbol{W}\Sigma\boldsymbol{W}^\top\boldsymbol{J}^\top, \tag{2}$$

where $\boldsymbol{W}$ and $\boldsymbol{J}$ denote the viewing transformation and the Jacobian of the affine approximation of perspective projection transformation (Zwicker et al., 2002), respectively. The 2D means are calculated through the projection matrix. After this, the pixel color is composited through the alpha blending of $N$ ordered 2D Gaussians:

$$\mathcal{C} = \sum_{i \in N} T_i \alpha_i \boldsymbol{c}_i \text{ with } T_i = \prod_{j=1}^{i-1} (1 - \alpha_i). \tag{3}$$

Here, $\alpha$ is obtained by multiplying the opacity $\boldsymbol{o}$ with the 2D covariance's probability computed from $\Sigma'$ and pixel coordinate on the image space.

### 3.2 MESH EXTRACTION

Our method can achieve high-quality editing using guided meshes obtained from various methods. In this section, we investigate different mesh extraction and reconstruction techniques with different mesh accuracy and processing time to guide the 3D Gaussians in our approach.

**Marching Cube for Gaussian Splatting.** In DreamGaussian (Tang et al., 2023a), the method attempts to summarize the alpha values of neighboring Gaussian points as the composite density value of marching cube sampling points. However, We found that this method often ignores the thin and small structures. With our Mesh-Gaussian binding strategy, we can achieve high-fidelity rendering with the inaccurate mesh and support smooth Gaussian manipulation.

**Screened Poisson Reconstruction.** 3D Gaussian Splatting could be considered a type of point cloud, making it intuitive to extract the mesh using the Poisson-reconstruction algorithm. However, the 3D Gaussians do not have normal vectors for reconstruction. Inspired by recent 3DGS inverse rendering methods (Gao et al., 2023; Liang et al., 2023), we allocate an additional gaussian attribute, normal $\boldsymbol{n}$, for 3D Gaussians, which is supervised by the pseudo normal derived from depth map. After training 3DGS with normal attributes, we can extract the mesh using the Screened poisson surface reconstruction (Kazhdan & Hoppe, 2013) algorithm.

**Neural Implicit Surfaces.** In this work, we also try to extract high-quality surfaces from the implicit representation utilizing the method proposed in NeuS (Wang et al., 2021). NeuS mesh has a large number of triangles, which negatively affects both training and inference speeds. We utilize mesh decimation techniques to reduce the count of triangles to approximately 300K.

### 3.3 BINDING GAUSSIAN SPLATTING ON MESH

Owing to the exceptional proficiency in high-fidelity and fast rendering, 3DGS has gained significant attention in differential rendering. However, despite being an explicit 3D representation, it currently lacks a method for effectively manipulating 3DGS while preserving high-quality rendering simultaneously. Mesh editing techniques, such as large-scale deformation, localized manipulation, and simulation, have been widely acknowledged and extensively researched for many years. Our primary objective is to associate the 3DGS with mesh triangles, enabling the manipulation of 3DGS and its rendering results following mesh editing.

Given a reconstructed or extracted triangular mesh $\boldsymbol{T}$ with $K$ vertices $\{\boldsymbol{v}_i\}_{i=1}^K$ and $M$ triangles $\{\boldsymbol{f}_i\}_{i=1}^M$, the goal of our method is to construct a 3DGS model bound to mesh triangles and optimize each Gaussian attribute $\{\boldsymbol{\mu}_i, \boldsymbol{q}_i, \boldsymbol{s}_i, o_i, \boldsymbol{c}_i\}$. To simplify the notation, we will omit the subscript in subsequent sections.

For each triangle $\boldsymbol{f}$ in the given mesh $\boldsymbol{T}$, which is composed of three vertices $(\boldsymbol{v_1}, \boldsymbol{v_2}, \boldsymbol{v_3})$, we initialize $N$ Gaussians on this triangle. To be specific, the mean position $\boldsymbol{\mu}$ of initialized Gaussians is formulated as $\boldsymbol{\mu} = (w_1 \boldsymbol{v_1} + w_2 \boldsymbol{v_2} + w_3 \boldsymbol{v_3})$, $\boldsymbol{w} = (w_1, w_2, w_3)$ is the pre-defined barycentric coordinate of each Gaussians attached on the triangle. And $\boldsymbol{w}$ satisfy $(w_1 + w_2 + w_3) = 1$.

**Gaussians on Mesh.** To achieve controllable 3DGS manipulation through the mesh, an intuitive way is to perfectly attach the GS to the triangle, as shown in the SuGaR (Guédon & Lepetit, 2023). With the rotation matrix denoted as $\boldsymbol{R} = \{\boldsymbol{r_1}, \boldsymbol{r_2}, \boldsymbol{r_3}\}$ and the scaling vector represented by $\boldsymbol{s} = (s_1, s_2, s_3)$, SuGaR train 3DGS with a flat Gaussian distribution on the mesh by setting $s_1 = \epsilon$, where $\epsilon$ is close to zero. $\boldsymbol{r_1}$ is defined by the normal vector $\boldsymbol{n}$ of the attached triangle. The Gaussians have only 2 learnable scaling factors $(s_2, s_3)$ instead of 3, and only 1 learnable 2D rotation rather

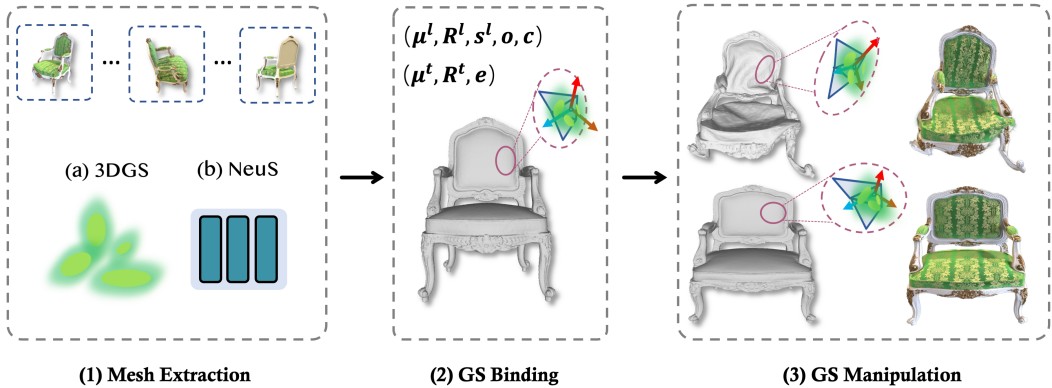

Figure 2: Overview of our method.(1) Firstly, we extract a triangular mesh from 3DGS (Kerbl et al., 2023) or a neural surface field(NeuS (Wang et al., 2021)). (2) Next, we bind $N$ Gaussians to each triangle in the local triangle space, and optimize the local Gaussian attributes ($\boldsymbol{\mu^l}, \boldsymbol{R^l}, \boldsymbol{s^l}, \boldsymbol{o}, \boldsymbol{c}$). The triangle attributes ($\boldsymbol{\mu^t}, \boldsymbol{R^t}, \boldsymbol{e}$) are calculated based on the triangle vertices. (3) Finally, we manipulate 3DGS by transferring the mesh manipulation directly, thus achieving manipulable rendering.

than a quaternion, to keep the Gaussians flat and aligned with the mesh triangles. This type of binding strategy heavily relies on mesh accuracy, which lacks the flexibility of 3DGS to model complex object rendering. Moreover, the mesh quality obtained from SuGaR is significantly inferior compared to the ground truth and recent neural surface reconstruction results, which increases the difficulty of editing.

**Gaussians on Mesh with Offset.** To compensate for the inaccuracy of the extracted mesh, it would be better to add an offset $\Delta\boldsymbol{\mu}$ to the Gaussians 3D mean $\boldsymbol{\mu}$, which enables the Gaussians to move out of the attached triangle $\boldsymbol{f}$. Although it could improve the rendering quality of the reconstructed static object, it would result in noisy and unexpected rendering distortion in the manipulated object due to the mismatched localized relative position between Gaussians.

**Triangle Shape Aware Gaussian Binding and Adapting.** To preserve the high-fidelity rendering results after manipulation, the key lies in maintaining the local rigidity and preserving the relative location between Gaussians, both for 3D means and rotations. Our key insight is to define a local coordinate system in each triangle space.

The first axis direction of triangle space is defined as the direction of the first edge. The second axis direction of triangle space is defined as the triangle's normal direction. The third axis direction of triangle space is defined as the cross product of the first and second axis. Then the triangle coordinate system rotation can be formulated as:

$$\boldsymbol{R^t} = [\boldsymbol{r_1^t}, \boldsymbol{r_2^t}, \boldsymbol{r_3^t}] = [\frac{(\boldsymbol{v_2} - \boldsymbol{v_1})}{\|\boldsymbol{v_2} - \boldsymbol{v_1}\|}, \boldsymbol{n^t}, \frac{(\boldsymbol{v_2} - \boldsymbol{v_1})}{\|\boldsymbol{v_2} - \boldsymbol{v_1}\|} \times \boldsymbol{n^t}] \tag{4}$$

where $\boldsymbol{v_1}, \boldsymbol{v_2}$ is the first and second vertex location repectively, $\boldsymbol{n^t}$ is the normal vector which can be calulated by:

$$\boldsymbol{n^t} = \frac{(\boldsymbol{v_2} - \boldsymbol{v_1}) \times (\boldsymbol{v_3} - \boldsymbol{v_1})}{\|(\boldsymbol{v_2} - \boldsymbol{v_1}) \times (\boldsymbol{v_3} - \boldsymbol{v_1})\|}. \tag{5}$$

We then optimize the Gaussians' local position $\boldsymbol{\mu^l}$ and local rotation $\boldsymbol{R^l}$ in triangle space instead of the global position and rotation in the original 3DGS.

Then the global rotation, scale and location of 3DGS are as follows:

$$\boldsymbol{R} = \boldsymbol{R^t}\boldsymbol{R^l}, \boldsymbol{s} = \boldsymbol{s^l}, \quad \boldsymbol{\mu} = \boldsymbol{R^t}\boldsymbol{\mu^l} + \boldsymbol{\mu^t} \tag{6}$$

where, $\boldsymbol{\mu^t}$ is the global coordinate of each triangle center. In practice, we initialize $N$ local Gaussian points and bind them for each Gaussian point, whose initialized position is on the triangle uniformly.

This Gaussian mesh binding method can preserve the relative position and rotation between Gaussians that are bonded on neighboring triangles after mesh manipulation. However, following mesh manipulation, not only does the triangle center change but also the triangle shape. With the altered triangle shape, the local Gaussian position and scaling should adjust accordingly. When the triangle enlarges, it is intuitive that the local scaling and position should expand as well:

$$\boldsymbol{R} = \boldsymbol{R^t R^l}, \boldsymbol{s} = \beta e \boldsymbol{s^l}, \quad \boldsymbol{\mu} = e \boldsymbol{R^t \mu^l} + \boldsymbol{\mu^t}, \tag{7}$$

where $\beta$ is a hyper-parameter, *adaption vector* $\boldsymbol{e} = [e_1, e_2, e_3]$ is designed to make sure that the global scaling $\boldsymbol{s}$ is proportionable to the triangle shape. The first axis is along the first edge, so $e_1$ is designed as the length $l_1$ of the first edge of the triangle. The second axis is along the normal direction, we set $e_2 = (0.5 * (e_1 + e_3))$. The third axis is perpendicular to the first edge, we set $e_3$ as the average length of the second and third edges $(0.5 * (l_2 + l_3))$.

### 3.4 MANIPULATE GAUSSIAN SPLATTING THROUGH MESH

Utilizing our triangle shape aware gaussian binding and adapting strategy, upon the completion of model training and mesh manipulation, the 3DGS is instantly manipulated and adapted. During mesh manipulation, the attributes in the local triangle space remain unchanged. The triangle rotation, position, and edge length can be calculated instantly. Therefore, the global Gaussian position, scaling, and rotation can be self-adaptively adjusted following our proposed formula. In this paper, we exhibit the 3DGS manipulation rendering outcomes, such as large-scale deformation, local manipulation, and soft-body simulation, which are driven by the manipulated mesh. Many 3D design software applications possess the capability to execute mesh manipulation. In our experiments, we employ Blender to manipulate the mesh.

## 4 EXPERIMENTS

### 4.1 TRAINING DETAILS

The first stage of our methods includes a mesh extracting stage, during which we extract triangular mesh from NeuS (Wang et al., 2021) or 3DGS (Screened Poisson or Marching Cube). However, the extracted mesh always contains enormous triangles, which we try to decimate to around 300K.

With the extracted mesh, we conduct the *triangle shape aware Gaussian binding and adapting strategy* on the mesh. For each triangle, we bind $N = 3$ Gaussian on the surface initially. The Gaussian attributes are optimized subsequently with the supervision of multi-view rendering loss in the second stage. We train our model for 30K iterations in the initial stage to extract mesh and 20K iterations in the second stage. All experiments are conducted on a single NVIDIA A100 GPU.

### 4.2 DATASETS, METRICS AND METHODS FOR COMPARISON

To evaluate our methods, we compare Mani-GS with previous editable novel view synthesis methods, a NeRF-based editing method NeRF-Editing (Liu et al., 2021) and a 3DGS-based editing method SuGaR (Guédon & Lepetit, 2023). For the evaluation, we employ the commonly used metrics:*PSNR*, *SSIM*, *LPIPS*. We evaluate our methods mainly on the NeRF Synthetic dataset (Mildenhall et al., 2021) and DTU dataset (Jensen et al., 2014)(in Appendix 9).

### 4.3 EVALUATION

**Static Rendering** Table 1 provides a quantitative comparison of all NeRF Synthetic 8 cases between our method and competing methods. We conducted experiments using their official code repository.

The numerical results of the NeRF-Editing were not presented in their original paper, we ran their code to provide more visual and numerical comparisons. As depicted in Table 1, we observed some outliers in *"Drums, Ficus"* that were lower than even 10 PSNR compared to ours and SuGaR. Therefore, we remove these outliers with a strikeout in the table. As can be observed, our approach surpasses all the baseline methods with respect to PSNR, SSIM, LPIPS, which means we achieve

Table 1: Quantitative comparison of our methods with NeRF-Editing (*N.E.*) (Liu et al., 2021) and *SuGaR* (Guédon & Lepetit, 2023) on NeRF Synthetic dataset in terms of SSIM, PSNR, LPIPS. (↑ means higher is better, ↓ means lower is better.) The best results are marked in bold.

| Subject | PSNR↑ | | | SSIM↑ | | | LPIPS↓ | | |
|---|---|---|---|---|---|---|---|---|---|
| | *N.E.* | *SuGaR* | Ours | *N.E.* | *SuGaR* | Ours | *N.E.* | *SuGaR* | Ours |
| Chair | 28.15 | 31.33 | **35.38** | 0.943 | 0.977 | **0.986** | 0.061 | 0.027 | **0.011** |
| Drums | 21.14 | 25.36 | **26.19** | 0.884 | 0.939 | **0.953** | 0.12 | 0.062 | **0.039** |
| Ficus | 23.82 | 29.94 | **35.40** | 0.909 | 0.959 | **0.986** | 0.101 | 0.039 | **0.013** |
| Hotdog | 32.67 | 35.45 | **37.49** | 0.969 | 0.980 | **0.984** | 0.048 | 0.035 | **0.019** |
| Lego | 29.16 | 32.09 | **36.33** | 0.944 | 0.968 | **0.982** | 0.074 | 0.037 | **0.015** |
| Material | 29.48 | 28.7 | **29.91** | 0.944 | 0.937 | **0.956** | 0.063 | 0.076 | **0.046** |
| Mic | 29.60 | 34.07 | **37.46** | 0.952 | 0.980 | **0.992** | 0.046 | 0.029 | **0.007** |
| Ship | 25.01 | 27.90 | **31.01** | 0.083 | 0.885 | **0.890** | 0.194 | 0.127 | **0.097** |
| Average | - | 30.61 | **33.65** | - | 0.9531 | **0.966** | - | 0.054 | **0.030** |

the best rendering quality. We are 3.0 higher than *SuGaR* in PSNR, 0.013 higher in SSIM, and 0.024 lower in LPIPS.

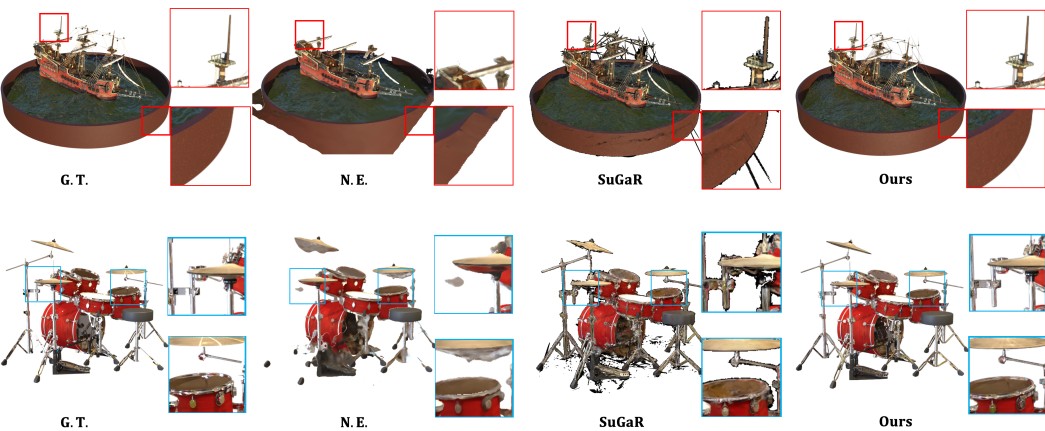

Figure 3: Visual comparison between ours, NeRF-Editing (Liu et al., 2021)(N.E.) and SuGaR (Guédon & Lepetit, 2023) for static rendering. It illustrates our proposed method can contain a much more accurate boundary in *"Ship"* , and detailed results in *"Drums"*.

In Figure 3, we present qualitative results of our approach and other methods in overview and zoom-in details. Our methods render more detailed and more accurate boundaries. For SuGaR, it attempts to bind 3D Gaussians on the triangle and enforce that the attached Gaussian is closely aligned on the triangle. In practice, they set the scale $s_1$ along the triangle's normal direction to a value close to zero, i.e., $s_1 = \epsilon$. This binding strategy heavily depends on the accuracy of the mesh. As can be observed in the third column of Figure 3, wrong geometry leads to an inaccurate rendering, especially in the boundary region.

**Manipulation Rendering** In Figure 4, we showcase our manipulation results. In these four cases, we manipulate the underlying mesh with large deformation, the *Chair* is stretched, *Lego* is tapered, *Ficus* and *Mic* is bent respectively. As demonstrated in *Chair, Ficus*, we have more accurate boundary and shape, as well as in the bottom region. This indicates that when the geometry is not that accurate, SuGaR can not adapt to compensate for geometry error, which results in missing geometry and dilated boundary geometry. For *Lego, Mic*, we can maintain the high rendering quality even after the large deformation, while SuGaR shows some distortion and noise in rendering results after large deformation. We did not present the manipulation results of NeRF-Editing (Liu et al., 2021) since we can not obtain their reasonable results using their code. So we compare with NeuMesh (Yang et al., 2022). As observed, we present more abundant and distinct details compared to NeuMesh. Please see the numerical comparison with NeuMesh in Appendix on DTU and NeRF datasets.

In addition to the large deformation, our method also produces promising results for local manipulation and physics simulation. Here we show an example of soft body simulation. In Figure 5 row

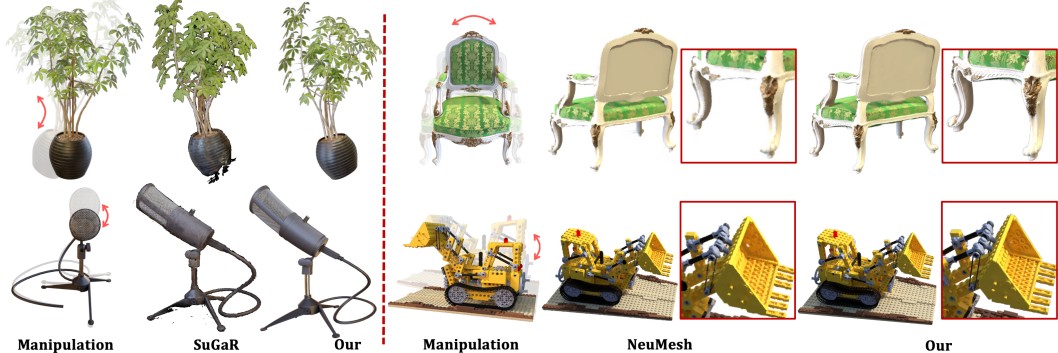

Figure 4: We offer an editing comparison between our method, SuGaR, and NeuMesh. Our approach demonstrates fewer artifacts and less blurring effects than SuGaR, and presents more abundant and distinct details compared to NeuMesh. For further details, please zoom in.

1, we try to *blend* the red sauce and yellow sauce of *Hotdog* as shown in the blue box, which shows satisfying editing and reasonable rendering quality. In *Drums*, we *repose* a cymbal and *elastically deform* a cymbal as shown in the blue box. After reposing and elastic deformation, we still preserve the photo-realistic rendering results. Note that the manipulation is achieved by manipulating the triangular mesh directly, the 3DGS rendering is achieved simultaneously with self-adaption.

In the second row of Figure 5, we present the rendering results of soft body simulation at different timesteps. As observed, we can achieve soft body simulation by just transferring the mesh simulation to 3DGS, which eliminates the need for a soft body simulation algorithm dedicated to 3DGS.

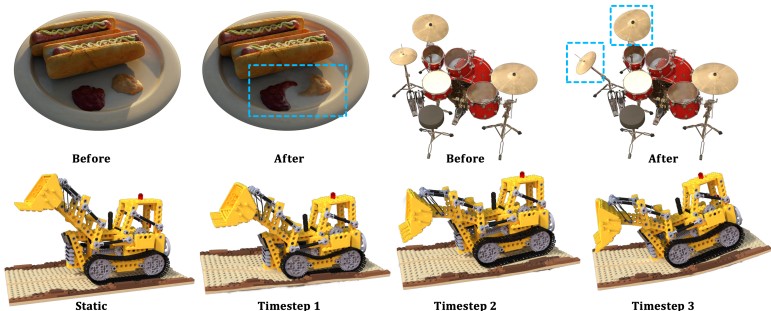

Figure 5: Visual results of *local manipulation* and *soft body simulation*. In row 1, we demonstrate that our proposed GS-mesh binding method enables us to support local part manipulation, In row 2, we showcase a 3DGS soft body simulation demo at different timesteps.

## 4.4 ABLATION STUDY

We conduct ablation studies to verify the effectiveness of triangle shape aware Gaussian binding and adapting method. We first evaluate the strategy of directly binding 3DGS to the mesh, which implies that the 3D position is fixed on the triangle. As shown in Table 2, the performance significantly drops, with a decrease of approximately 2.6 PSNR compared to our best model. For visual ablation in Figure 6, *3DGS on NeuS Mesh* after deformation shows a boundary with many burrs.

Next, we verify the effectiveness of adding 3D offset for 3DGS on Mesh. Although the offset can enhance the fitting of 3DGS to the static scene, as demonstrated in Table 2, it fails to generate satisfactory deformation rendering results because the offset only fits the static scene and remains unchanged during subsequent deformations. Consequently, it leads to significant noise and distortion after manipulation in Table 2.

Finally, we conduct experiments with different meshes extracted using different methods. The 3DGS Marching Cube Mesh (**3DGS MC**) is of low quality, including a dilated boundary and very noisy surface, as can be observed in Figure 6 row 2. The screened poisson mesh (**Poisson Recon.**) has

Table 2: Quantitative ablation comparison between *3DGS On NeuS Mesh*, *NeuS Mesh + Offset*, *Ours with Marching Cube Mesh*,*Ours with Screened Poisson Mesh* on NeRF Synthetic dataset. (↑ means higher is better, ↓ means lower is better.)

| Method | PSNR↑ | SSIM↑ | LPIPS↓ |
|---|---|---|---|
| 3DGS On NeuS Mesh | 30.87 | 0.9521 | 0.0447 |
| NeuS Mesh + Offset | 32.48 | 0.9625 | 0.0341 |
| Ours + Marching Cube Mesh | 32.11 | 0.9602 | 0.035 |
| Ours + Screened Poisson Mesh | 33.42 | 0.9638 | 0.0324 |
| Ours + NeuS Mesh | **33.45** | **0.9646** | **0.0309** |

some unconnected regions and missing parts compared with NeuS mesh (**NeuS**). However, using our triangle shape aware Gaussian binding and adapting method can still achieve 3DGS manipulation and maintain high-fidelity rendering even after very large deformation as shown in Figure 6 row 2. As shown in Table 2, the numerical results obtained with screened poisson mesh are only slightly lower than those obtained with NeuS mesh. When the mesh is of low quality, such as the **MC mesh**, the quantitative results are approximately 1 PSNR lower than the best, but still 2 PSNR higher than only binding 3DGS on the best Mesh (NeuS mesh), and 1.5 PSNR higher than SuGaR.

We also evaluate our method using mesh extracted from SuGaR, whose mesh is extremely dense. The PSNR using SuGaR mesh is 33.67, which is 3 PSNR higher than the original SuGaR results.

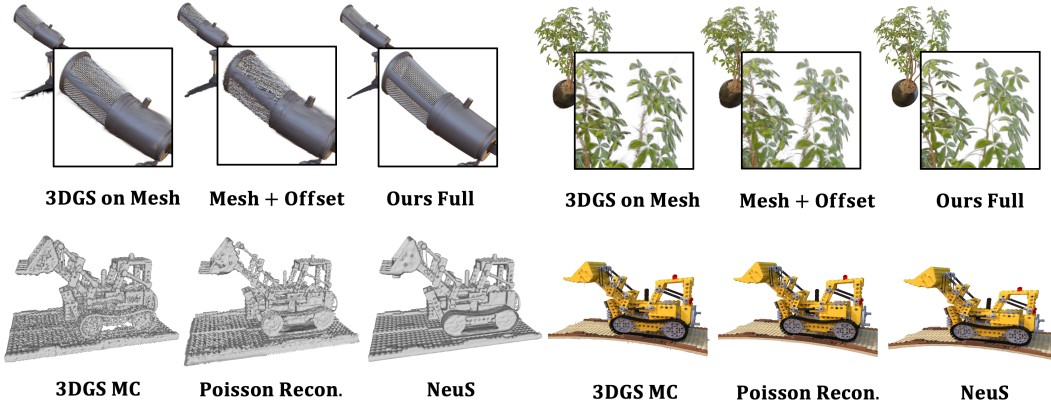

Figure 6: Visual Results of Ablation Study. After deformation, (**3DGS on Mesh**) shows a burring boundary, (**Mesh + Offset**) leads to significant noise and distortion, (**Ours Full**) can maintain the high fidelity rendering. In the second row, we demonstrate that even with a low-quality mesh, we can still achieve high-quality editable rendering.

## 5 CONCLUSION AND LIMITATION

In this paper, we introduce a triangle shape aware Gaussian binding strategy with self-adaptation, which supports various 3DGS manipulations, maintains rendering quality, and has a high tolerance for mesh accuracy. We evaluate our methods on the NeRF synthetic dataset and demonstrate state-of-the-art results, showcasing various 3DGS manipulations, including large deformations, local manipulations, and soft body simulations.

During our experiments, we noticed that some results still exhibit distortions. When the local region of the manipulated mesh contains highly non-rigid deformations, it can result in rendering distortions. Additionally, during our simulation demos, we found that conducting physics simulations on meshes with more than 35K triangles can take hours. It would be a novel direction to explore fast simulation methods for 3DGS. Finally, we found that our results may not have accurate boundary rendering when the extracted mesh has a significant discrepancy from the ground truth mesh, such as unconnected regions.

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

## A APPENDIX

In this appendix, we provide more visual rendering and geometry results on the NeRF (Mildenhall et al., 2021) Synthetic dataset including some video demos (included in supply.) in Sec. A.1. And we further evaluate our methods on the DTU (Jensen et al., 2014) dataset and provide qualitative and quantitative results in Sec. A.2. In addition, we give an efficiency analysis of training time and inference time in Sec. A.3. Finally, we describe more implementation details of our method in Sec. A.4.

### A.1 MORE RESULTS ON NERF SYNTHETIC DATASET

**Soft Body Simulation.** In addition to the visual results presented in Figure 5 of the main paper, we also provide the geometry after simulation and rendering at different viewpoints in Figure 7. To improve the speed of the mesh simulation, we decimated the original mesh from 300K to 35K triangles. While this may result in some decrease in rendering quality due to the reduced number of triangles as well as Gaussians, it was necessary to ensure reasonable simulation speed.

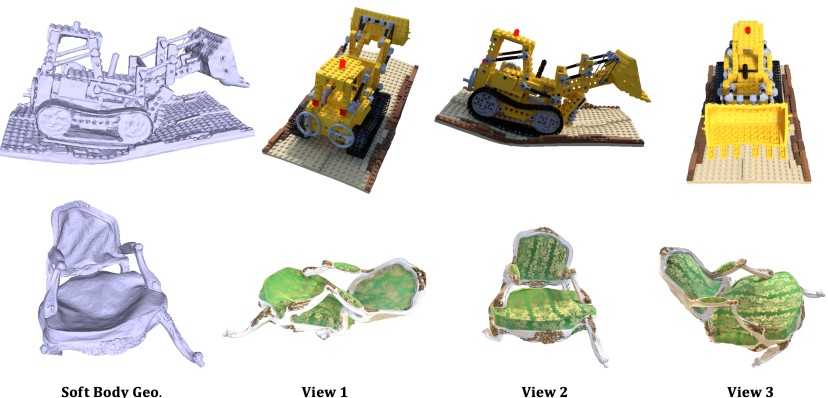

| Soft Body Geo. | View 1 | View 2 | View 3 |

Figure 7: Visual results of softbody simulation at different viewpoints. The left column displays the geometry after simulation, while the right three columns showcase the rendering results from three different viewpoints.

**Demo Video.** In order to further demonstrate the effectiveness of our methods, we have provided additional visual videos showcasing large deformation, soft body simulation, and local manipulation. These videos can be accessed through a local webpage by navigating to the *mani-gs* folder and clicking on *index.html*.

| Methods | DTU | | | NeRF 360° Synthetic | | |
|---|---|---|---|---|---|---|
| | PSNR ↑ | SSIM ↑ | LPIPS ↓ | PSNR ↑ | SSIM ↑ | LPIPS ↓ |
| NeuS (Wang et al., 2021) | 26.352 | 0.909 | 0.176 | 30.588 | 0.960 | 0.058 |
| Neu-Mesh (Yang et al., 2022) | 28.289 | 0.921 | 0.117 | 30.95 | 0.951 | 0.043 |
| Ours | **31.496** | **0.943** | **0.088** | **36.67** | **0.986** | **0.013** |

Table 3: We compare quantitative rendering quality with NeuS (Wang et al., 2021) and NeuMesh (Yang et al., 2022) on the DTU dataset and the NeRF Synthetic dataset.

**Numerical Comparison with NeuS and Neu-Mesh.** We have compared our methods with NeuS and Neu-Mesh on the NeRF Synthetic dataset, and the results are presented in Table 3. According

to NeuMesh, they only chose 4 representative scenes (Lego, Mic, Chair, Hotdog) that worked well for evaluation. Therefore, we use the same 4 scenes for comparison. Our approach achieves a PSNR score that is 5.7 higher than Neu-Mesh, an SSIM score that is 0.035 higher than Neu-Mesh, and an LPIPS score that is 0.03 lower than Neu-Mesh. These results demonstrate that our approach has achieved the best overall performance in all metrics compared to neural implicit field methods (NeuS and Neu-Mesh) on the NeRF Synthetic dataset.

**Binding Gaussians on low-quality mesh.** In some extreme cases, the Screend Poisson surface reconstruction method may result in a very low-quality triangular mesh. However, with our mesh-Gaussians binding strategy, we can still generate much better rendered images than those produced by *SuGaR* (Guédon & Lepetit, 2023), even though *SuGaR* may have a much better mesh in such case as shown in Figure 8.

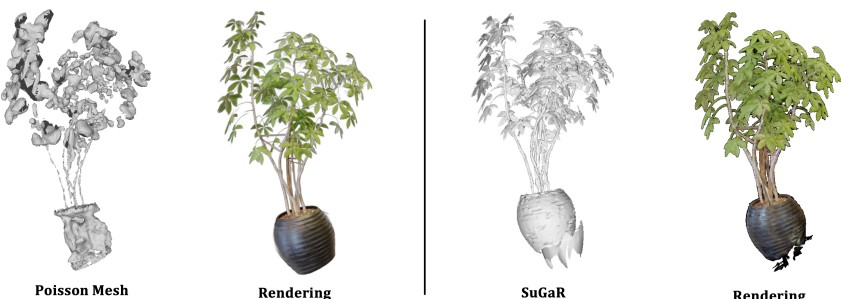

Poisson Mesh      Rendering      SuGaR      Rendering

Figure 8: Binding Gaussians on a low-quality mesh (ours on the left, *SuGaR* on the right), we are still able to achieve high-fidelity manipulated rendering results when the mesh we generate of Screened Poisson reconstruction is of low quality. In contrast, *SuGaR* fails to produce satisfactory results, even though it has a better mesh in this particular case.

## A.2 More Results on DTU Dataset

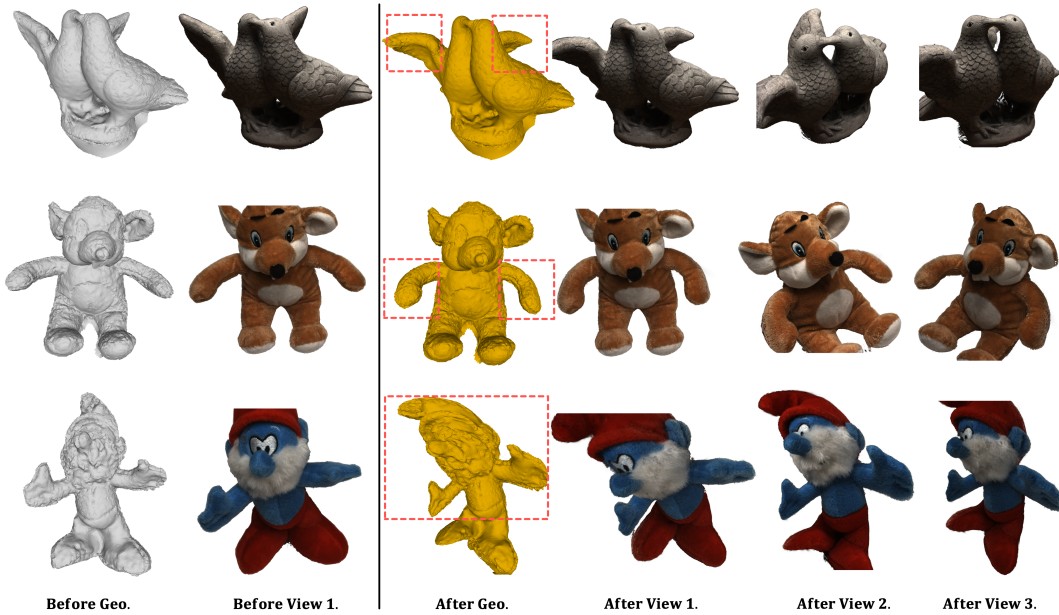

Before Geo.      Before View 1.      After Geo.      After View 1.      After View 2.      After View 3.

Figure 9: Manipulation rendering results in DTU dataset. The left two columns showcase the geometry and rendered image before manipulation, while the right three columns showcase the geometry and rendered image after manipulation. To highlight the deformed area, we have enclosed it within a red rectangle.

We have also evaluated our approach on the DTU dataset, which includes 15 cases with multi-view images as input. The results are presented in Table 3, and demonstrate that our approach achieves a PSNR score that is 2.8 higher than Neu-Mesh, and a SSIM score that is 0.022 higher than Neu-Mesh. Additionally, our approach has a LPIPS score that is 0.016 lower than Neu-Mesh, indicating that our approach achieves the best overall performance on the DTU dataset.

We have also presented the manipulation results in Figure 9. The left two columns showcase the geometry and rendered image before manipulation, while the right three columns showcase the geometry and rendered image after manipulation. In the first row, the wing of a pigeon is manipulated, while in the second row, the arms of a tiger are swinging down. In the third row, a toy is twisted to the left. The manipulation results presented in Figure 9 demonstrate that our approach can successfully transfer mesh manipulation to Gaussian-Splatting, resulting in accurate and visually appealing results.

## A.3 EFFICIENCY ANALYSIS

The efficiency of GS Binding training and rendering speed depends on the number of Gaussians, which is the product of the triangle number $T$ and the Gaussians number for each triangle $N$. In Table 4, We first fixed $T$ and tested different values of $N$. Our results indicate that $N=3$ leads to the best rendering quality while keeping a competitive rendering speed. When $N = 1$, the PSNR slightly decreased with a faster training and rendering speed.

We also evaluated the impact of underlying mesh resolution by testing meshes with different triangles (*270K, 150K, 70K*). As shown in Table 4, the rendering quality decreases while efficiency improves with decreasing mesh resolution.

Regarding the editing time, it primarily depends on the time cost of mesh editing. We use Blender for mesh editing, and in our experience, *local manipulation* and *large deformation* can be achieved instantly. *Soft body simulation* can be a more time-consuming process, as it depends on the simulation algorithm employed in Blender.

Table 4: Efficiency Analysis

|  | N=4 | N=3 | N=1 | N=1 | N=1 |
|---|---|---|---|---|---|
| *Triangles(K)* | 270 | 270 | 270 | 150 | 70 |
| *Points(K)* | 1080 | 810 | 270 | 150 | 70 |
| *Training (min)* | 16 | 13 | **7** | 5.5 | 4.5 |
| *Speed (FPS)* | 244 | 300 | **452** | 571 | 572 |
| *PSNR* | 36.36 | **36.39** | 36.27 | 35.86 | 34.52 |

## A.4 IMPLEMENTATION DETAILS

### A.4.1 TRAINING DETAILS OF MESH EXTRACTION STAGE

As outlined in our main paper, the first stage of our approach involves mesh extraction. While we utilize the NeuS (Wang et al., 2021) mesh as the foundation for binding Gaussians, we also explore extracting mesh from Gaussian-Splatting.

In this work, we try to extract triangular mesh using the Screened Poisson surface reconstruction (Kazhdan & Hoppe, 2013) method from trained Gaussian-Splatting model. We incorporate a normal attribute $n$ for each 3D Gaussian and optimize the normal attribute with the pseudo-normal constraint.

The normal consistency is quantified as follows:

$$\mathcal{L}_n = \|\mathcal{N} - \tilde{\mathcal{N}}\|_2. \tag{8}$$

where $\mathcal{N}$ is the rendered-normal map, $\tilde{\mathcal{N}}$ is the pseudo-normal map computed from rendered depth map.

Besides the normal constraint $\mathcal{L}_n$, the ordinary L1 Loss and Structural Similarity Index (SSIM) loss are also incorporated into optimization by comparing the rendered image $\mathcal{C}$ with the observed image

$\mathcal{C}_{gt}$. To address the issue of unwarranted 3D Gaussians in the background region, we employ a mask cross-entropy loss. This loss is defined as follows:

$$\mathcal{L}_{mask} = -B^m \log B - (1 - B^m) \log (1 - B), \tag{9}$$

where $B^m$ denotes the object mask and $B$ denotes the accumulated transmittance $B = \sum_{i \in N} T_i \alpha_i$.

Then all the loss terms can be summarized as follows:

$$\mathcal{L}_{stage1} = \lambda_1 \mathcal{L}_1 + \lambda_2 \mathcal{L}_{SSIM} + \lambda_3 \mathcal{L}_n + \lambda_4 \mathcal{L}_{mask}, \tag{10}$$

where $\lambda_1 = 1, \lambda_2 = 0.2, \lambda_3 = 0.01, \lambda_4 = 0.1$. We train this stage for 30K steps with adaptive density control, which is executed at every 500 iterations within the specified range from iteration 500 to 10K. Once the training stage is complete, we proceed with Screened Poisson surface reconstruction using the positions and normals of the Gaussians as input. The mesh extraction process takes less than 1 minute to complete.

In addition to mesh extraction, we also utilize Gaussian-Splatting Marching-Cube to extract the triangular mesh. Our approach involves sampling a grid with a resolution of $256 \times 256 \times 256$. For each sampling point, we identify its nearest Gaussian points. Sampling points that have the nearest Gaussians within a pre-defined distance threshold $\tau$ are assigned a density value of 1, while those that do not meet the threshold are assigned a density value of 0. $\tau$ is set to 0.01 in practice. The density threshold for Marching-Cube is set to 1e-4.

Based on the visual comparison, the overall mesh quality can be ranked as follows: NeuS > Poisson Reconstruction > Marching-Cube.

### A.4.2 TRAINING DETAILS OF GAUSSIAN-BINDING STAGE

To ensure an accurate representation of each triangle, we bind $N$ Gaussians to it. Prior to training, we initialize the positions of the Gaussians on the attached triangle. The $N$ initialized position is calculated using a barycentric coordinate, with a predefined barycentric coordinate set of $[1/2, 1/4, 1/4]$, $[1/4, 1/2, 1/4]$, $[1/4, 1/4, 1/2]$. For the hyper-parameter $\beta$ mentioned in main paper equation (8), we set $\beta = 10$ in most cases, $\beta = 100$ in *Materials*.

In the Gaussian-Binding stage, we don't perform adaptive control because we find it doesn't influence the final performance. We also train 300K iterations in this stage with L1 loss, SSIM loss and mask entropy loss. The overall loss in this can be summarized as follows:

$$\mathcal{L}_{stage2} = \lambda_1 \mathcal{L}_1 + \lambda_2 \mathcal{L}_{SSIM} + \lambda_3 \mathcal{L}_{mask}, \tag{11}$$

where $\lambda_1 = 1, \lambda_2 = 0.2, \lambda_3 = 0.1$.

