# OpenReview forum: "Mani-GS: Gaussian Splatting Manipulation with Triangular Mesh"
_ICLR.cc/2025/Conference — ICLR 2025 Conference Withdrawn Submission_

### Official Review · Reviewer_6rXy · 2024-10-29

**Soundness:** 2
**Presentation:** 2
**Contribution:** 2
**Rating:** 5
**Confidence:** 5

**Summary:**

This article focuses on the editing of Gaussians. Its central idea is to first perform mesh modeling of the scene, then bind 3DGS to the mesh for topology-consistent training. To better bind Gaussians to the mesh’s triangular surfaces, the paper proposes a coordinate system definition method based on triangles, allowing the topology to maintain a more stable structure. Once completed, this enables Gaussian editing and simulation similar to mesh manipulation. The authors conducted experiments on NeRF synthetic data and DTU data, achieving the expected editing effects.

**Strengths:**

1. The article has a clear logic and provides an in-depth analysis of the problem. For example, when discussing how to bind Gaussians to the mesh (Sec.3.3), the authors compared two alternative methods ("Gaussians on Mesh" and "Gaussians on Mesh with Offset"), analyzing the principles, advantages, and disadvantages of each. Another example is the authors' discussion of the results from different mesh extraction methods (Sec.3.2).
2. The supplementary materials are meticulously prepared, and the demo presentation is impressive, showcasing excellent results on editing and simulating.

**Weaknesses:**

1. The discussion of some works is insufficient. For example, GaMeS and Mesh-GS are mentioned in the related work section, but as the most closely related and recent Gaussian methods, they are not included in the experimental comparisons. Methodologically, I feel that the Gaussian binding approach in this article is very similar to that of GaMeS, yet the authors do not discuss this point. The baselines the authors compare are outdated and are insufficient to demonstrate the superiority of their method.
2. The range of data types this article's method can be applied to is not diverse enough. From the authors' experiments, it currently only supports the editing of small objects and relies heavily on the topology mesh obtained from mesh reconstruction algorithms (e.g., NeuS). If the object becomes more complex or includes a complex background, this approach is likely to produce a degraded-quality mesh, making it impossible to proceed with subsequent binding operations.

**Questions:**

1. There are some citation errors, such as the reference to Mesh-GS (Waczynska et al., 2024), which actually pertains to the GaMeS paper.
2. Have you tried testing your method on real-world data with backgrounds, such as the LLFF dataset? How effective is it in such cases?

---

### Official Review · Reviewer_1SJC · 2024-10-29

**Soundness:** 3
**Presentation:** 3
**Contribution:** 2
**Rating:** 5
**Confidence:** 4

**Summary:**

This submission describes an approach for deforming a 3D scene represented by 3D Gaussians. Towards this goal, the proposed method extracts a triangle mesh, binds the 3D Gaussians representing the scene to the mesh (on- and off-surface), and then uses the mesh to drive rigid and non-rigid deformations of the 3D Gaussians for object deformation.

**Strengths:**

The paper is reasonably well written and easy to understand. It addresses the important challenge of editing 3D scenes represented by 3D Gaussians. The qualitative results look compelling and quantitatively outperform the sugar baseline.

**Weaknesses:**

There are several weaknesses:

1. The proposed methods is incremental compared with sugar. The proposed method is basically sugar, which binds the optimized 3D Gaussians to the mesh surface. with an additional offset. This seems like a simple extension. More advanced extensions of sugar already exists, including

Guedon and Lepetit, Gaussian Frosting: Editable Complex Radiance Fields with Real-Time Rendering, ECCV 2024

which model a much broader class of objects than both sugar and the proposed work.

2. The related work discussion is too focused on NeRF, instead of giving a more comprehensive snapshot of approaches that enable animatable / deformable 3D Gaussians. A few examples:

Huang and Yu, GSDeformer: Direct Cage-based Deformation for 3D Gaussian Splatting
Abdal et al., Gaussian Shell Maps for Efficient 3D Human Generation, CVPR 2024
Yang et al., Deformable 3D Gaussians for High-Fidelity Monocular Dynamic Scene Reconstruction, CVPR 2024

So relevant papers published at CVPR 2024, ECCV 2024, and also SIGGRAPH Asia 2024

3. Some claims on the capabilities of the proposed system seem exaggerated

The presented results look good, but they mainly show local and non-rigid deformations. I did not see examples that show the claimed "large deformations" (see g.g. abstract & introduction)

**Questions:**

1. Can you please summarize additional recent baselines, including Gaussian Frosting, and compare against those?

2. Please show large deformations or avoid claiming them.

---

### Official Review · Reviewer_umHH · 2024-10-31

**Soundness:** 3
**Presentation:** 3
**Contribution:** 1
**Rating:** 5
**Confidence:** 5

**Summary:**

This work proposes a method for 3D Gaussian manipulation using a mesh as a proxy. By defining a local coordinate system for each triangle, the paper associates Gaussians with each triangle in a self-adaptive manner. The paper is clearly illustrated and thoroughly demonstrated.

**Strengths:**

- The paper is well written, and the analysis is comprehensive.
- The idea of correlating the scale of 3D Gaussians with the shape of the triangles to better handle large-scale deformations is reasonable.
- The experimental results appear to be valid.

**Weaknesses:**

- 3D Gaussian Spatting achieves high-quality rendering results primarily due to its split/clone mechanism, which adaptively adjusts the number of points in the scene. However, this paper limits the number of Gaussians in each triangle face, which may restrict its fitting capability. Nevertheless, the rendering metrics in Table 1 appear to be very high, with some even exceeding those of the original 3DGS; this raises questions.
- The main innovation of this paper lies in the introduction of $e$ in Equation 7 to better handle large-scale deformations. However, this is not evident in the ablation study. In fact, both the 3DGS on Mesh and Mesh + Offset experiments seem not to address the rotation of Gaussians, which is unreasonable.
- The current experimental examples are focused on hard surfaces. However, the greater advantage of 3DGS compared to meshes lies in rendering scenes without well-defined surfaces. How does this method perform on fuzzy geometry (e.g., the data from "Adaptive Shells for Efficient Neural Radiance Field Rendering")?

**Questions:**

The training of 3DGS in the paper is conducted entirely in static scenes, which fails to effectively learn the correspondence that Gaussian and mesh should maintain during motion. If a 3D Gaussian is trained separately and then matched to the mesh surface (transforming coordinates from world space to the local space of each triangle), can good manipulation still be achieved?

---

### Official Review · Reviewer_XDBV · 2024-11-03

**Soundness:** 3
**Presentation:** 3
**Contribution:** 2
**Rating:** 5
**Confidence:** 5

**Summary:**

The paper utilizes a given triangular mesh for free-form shape deformation of Gaussian-splatting with self-adaption. By parameterizing each Gaussian in the local triangle space and decoupling the local and global transformations, the proposed method maintains the local rigidity and preserves the relative location between Gaussian, which is robust to inaccurate meshes. The authors demonstrate the method editability in three tasks, large deformation, local manipulation, and soft body simulation.

**Strengths:**

1. The paper is well written and straightforward.
2. The paper proposes to bind Gaussians to a local triangle space, which maintains the local rigidity and preserves the relative
location between Gaussians, allowing the method to preserve the high-fidelity rendering results after manipulation.
2. The manipulation results are vivid and interesting, especially the soft body simulation.
3. The authors demonstrate the editability of the method on three different tasks, which shows the capability of the method in various scenarios.

**Weaknesses:**

1. The authors need to compare their method with GaMeS or Mesh-GS to demonstrate their contributions. In comparison to GaMeS which constrains the Gaussians on the surface exactly, the main contributions of Mani-GS are (1) attaching the Guassians to local space rather than global space, and (2) allowing Gaussians to offset out of the attached triangle. Could the authors provide some qualitative results that support those two contributions? In terms of the rendering quality given an inaccurate mesh and the rendering quality after manipulation. For example, given the Poisson mesh in Fig.8, where part of the pot is missing, Mani-GS can better fill the missing part than GaMeS since it allows the offset. And for example, show a case where the rendering quality of Mani-GS is better than GaMeS after manipulation due to the local triangle space.

2. In line 333 authors propose to use an adaption vector to scale both the offset vector and the scale of the Gaussian. However, the adaption vector solely depends on the length of the three triangle edges. Imagine stretching a triangle along its plane, e1, e2, e3 will all increase as the edge lengths get larger. Since e2 increases, the offset of the Gaussian along the triangle's normal direction will get larger, the Gaussian will move farther from the plane. A concrete example could be the Poisson mesh in Fig. 8, since part of the pot is missing, to be able to reconstruct the pot there must be lots of Gaussians that have large offsets along the normal directions, in that case if you stretch the pot vertically, I'd expect the Gaussians to expand horizontally as well. Does this lead to artifacts empirically? I'm happy to hear any comments on this.

3. Is the mesh used in Table 1 extracted from SuGaR? If not what's the mesh used there and could you provide the results using the SuGaR mesh for a fair comparison? If yes seems the average PSNR is different from what is mentioned later in line 505.

4. Do you regularize the scale of the local position \mu? I'm concerned that a Gaussian could significantly offset the attached triangle, potentially causing artifacts after manipulation.

5. In line 44 NeRF-Editing is referred to as Yuan et al. 2022, but in the rest of the paper (for example lines 365 and 379) it becomes Liu et al. 2021. The former approach is more relevant for comparison, as it aligns more closely with the context of free-form shape deformation, which the latter approach does not directly address. Is it a typo?

**Questions:**

Please check the weakness section. My main concern is the comparison with GaMeS or Mesh-GS.

---

### Official Review · Reviewer_EcEp · 2024-11-03

**Soundness:** 3
**Presentation:** 1
**Contribution:** 2
**Rating:** 5
**Confidence:** 4

**Summary:**

The authors improve the methodology of manipulating renderings generated by 3D Gaussian Splatting (3DGS). To achieve this, they propose the use of triangular mesh (generated by NeuS) as initial input to the 3DGS. Additionally, the authors propose a triangle-aware 3DGS to improve the manipulation and rendering capability.

**Strengths:**

The strengths of this work rely mainly on addressing a highly relevant problem and achieving great values compared to their chosen state-of-the-art methods.

**Weaknesses:**

This work suffers from a few larger issues:

- Poor writing quality. Here, we mostly mean that the paper heavily introduces and talks about NeRF in the introduction and related work, while this is not relevant for understanding the paper. Further, structurally the paper needs some improvements (for example Figure is mentioned on page 4 but not seen until page 6)
- In general, while the method works decently, the contributions do not seem to be enough
- Compared to SuGaR, the author here uses better meshes that are generated from NeuS (higher training time). The authors should address the differences in the training in their work.

**Questions:**

- On line 505, you mention that the results using SuGaR mesh are 33.676 dB (You + SuGaR), which is higher than 33.34 (You + NeuS). Why use NeuS if this is the case? If this is the case, the contribution on a quantitative level does not seem to be significant.
- In Table 1, please add, if possible, the rendering results of NeuS so that it can be seen how much the authors improve on the work of NeuS.

---

### Note · Authors · 2024-11-13

I have read and agree with the venue's withdrawal policy on behalf of myself and my co-authors.